# Intensified screening for SARS-CoV-2 in 18 emergency departments in the Paris metropolitan area, France (DEPIST-COVID): A cluster-randomized, two-period, crossover trial

Judith Leblanc[1]*, Lisbeth Dusserre-Telmon[2], Anthony Chauvin[3], Tabassome Simon[4], Chiara E. Sabbatini[5], Karla Hemming[6], Vittoria Colizza[5], Laurence Bérard[7], Jérôme Convert[8], Sonia Lazazga[9], Carole Jegou[10], Nabila Taibi[11], Sandrine Dautheville[12], Damien Zaghia[13], Camille Gerlier[14], Muriel Domergue[15], Florine Larrouturou[16], Florence Bonnet[17], Arnaud Fontanet[18], Sarah Salhi[7], Jérôme LeGoff[19], Anne-Claude Crémieux[20], On behalf of the DEPIST-COVID group[¶], FHU IMPEC (Improving Emergency Care) group

1 Sorbonne Université, INSERM, Pierre Louis Institute of Epidemiology and Public Health; Assistance Publique-Hôpitaux de Paris (AP-HP), Hôpital St Antoine, Clinical Research Platform Paris-East, Paris, France, 2 Centre Hospitalier de Melun, Emergency department, Melun, France, 3 AP-HP, Hôpital Lariboisière, Emergency department; Université Paris Cité, INSERM U942 MASCOT, Paris, France, 4 AP-HP, Hôpital St Antoine, Clinical Research Platform Paris-East; Sorbonne Université, Department of Clinical Pharmacology, Paris, France, 5 Sorbonne Université, INSERM, Pierre Louis Institute of Epidemiology and Public Health, Paris, France, 6 University of Birmingham, Institute of Applied Health Research, Birmingham, United Kingdom, 7 AP-HP, Hôpital St Antoine, Clinical Research Platform Paris-East, Paris, France, 8 AP-HP, Hôpital Lariboisière, Emergency department, Paris, France, 9 Centre Hospitalier de Gonesse, Emergency department, Gonesse, France, 10 AP-HP, Hôpital Avicenne, Emergency department, Bobigny, France, 11 AP-HP, Hôpital Pitié-Salpêtrière, Emergency department, Paris, France, 12 AP-HP, Hôpital Tenon, Emergency department, Paris, France, 13 AP-HP, Hôpital Beaujon, Emergency department, Clichy, France, 14 Hôpital Paris St Joseph, Emergency department, Paris, France, 15 AP-HP, Hôpital Européen Georges Pompidou, Emergency department, Paris, France, 16 AP-HP, Hôpital Louis Mourier, Emergency department, Colombes, France, 17 AP-HP, Hôpital St Antoine, Emergency department, Paris, France, 18 Institut Pasteur, Emerging Diseases Epidemiology Unit; PACRI unit, Conservatoire National des Arts et Métiers, Paris, France, 19 Université Paris Cité, INSERM U976, INSIGHT Team; AP-HP, Hôpital St Louis, Virology Department, Paris, France, 20 AP-HP, Hôpital St Louis, Infectious Diseases Department; Université Paris Cité, FHU PROTHEE, Paris, France

☯ These authors contributed equally to this work.
¶ Membership of the DEPIST-COVID group is provided in the document Appendix M. Study group.
* judith.leblanc@aphp.fr

**Data Availability Statement:** The study data are owned by the Assistance Publique - Hôpitaux de Paris (AP-HP) sponsor, "Département de la Recherche Clinique et du Développement", and are not freely available. Requests to access the data

## Abstract

### Background

Asymptomatic and paucisymptomatic infections account for a substantial portion of Severe Acute Respiratory Syndrome Coronavirus 2 (SARS-CoV-2) transmissions. The value of intensified screening strategies, especially in emergency departments (EDs), in reaching asymptomatic and paucisymptomatic patients and helping to improve detection and reduce transmission has not been documented. The objective of this study was to evaluate in EDs whether an intensified SARS-CoV-2 screening strategy combining nurse-driven screening

should be directed to: DJENNAOUI Fatiha <fatiha.djennaoui@aphp.fr> and [DRC] Secretariat Promotion Délégation à la Recherche Clinique et à l'Innovation <drc-secretariat-promotion@aphp.fr>.

**Funding:** The study was funded by the Agence Nationale de Recherche sur le Sida et les Hépatites Virales | Maladies Infectieuses Emergentes (ANRS| MIE) (https://anrs.fr/fr/), Paris (JL) and by the Région Île-de-France (https://www.iledefrance.fr/), France (JL). The funders of the study had no role in study design, data collection and analysis, decision to publish or preparation of the manuscript.

**Competing interests:** I have read the journal's policy and the authors of this manuscript have the following competing interests: JLG reports personal fees for symposia presentations organized by Abbott Rapid Diagnosis SAS in 2021 and Qiagen Inc in 2022. TS reports grants or contracts from AstraZeneca, Bayer, Boehringer, Daiichi-Sankyo, Eli-Lilly, GSK, Novartis, Sanofi, payment or honoraria for lectures from Servier, Novartis, participation on a Data Safety Monitoring Board or Advisory Board from Ablative Solutions, Air Liquide, AstraZeneca, Sanofi, Novartis, 4Living Biotech.

**Abbreviations:** CI, confidence interval; COVID-19, Coronavirus Disease 2019; ED, emergency department; MN, Miettinen–Nurminen; NEAR, nicking and extension amplification reaction; PCR, polymerase chain reaction; RdRp, RNA-dependent RNA polymerase; SARS-CoV-2, Severe Acute Respiratory Syndrome Coronavirus 2.

for asymptomatic/paucisymptomatic patients with routine practice (intervention) could contribute to higher detection of SARS-CoV-2 infections compared to routine practice alone, including screening for symptomatic or hospitalized patients (control).

## Methods and findings

We conducted a cluster-randomized, two-period, crossover trial from February 2021 to May 2021 in 18 EDs in the Paris metropolitan area, France. All adults visiting the EDs were eligible. At the start of the first period, 18 EDs were randomized to the intervention or control strategy by balanced block randomization with stratification, with the alternative condition being applied in the second period. During the control period, routine screening for SARS-CoV-2 included screening for symptomatic or hospitalized patients. During the intervention period, in addition to routine screening practice, a questionnaire about risk exposure and symptoms and a SARS-CoV-2 screening test were offered by nurses to all remaining asymptomatic/paucisymptomatic patients. The primary outcome was the proportion of newly diagnosed SARS-CoV-2–positive patients among all adults visiting the 18 EDs. Primary analysis was by intention-to-treat. The primary outcome was analyzed using a generalized linear mixed model (Poisson distribution) with the center and center by period as random effects and the strategy (intervention versus control) and period (modeled as a weekly categorical variable) as fixed effects with additional adjustment for community incidence. During the intervention and control periods, 69,248 patients and 69,104 patients, respectively, were included for a total of 138,352 patients. Patients had a median age of 45.0 years [31.0, 63.0], and women represented 45.7% of the patients. During the intervention period, 6,332 asymptomatic/paucisymptomatic patients completed the questionnaire; 4,283 were screened for SARS-CoV-2 by nurses, leading to 224 new SARS-CoV-2 diagnoses. A total of 1,859 patients versus 2,084 patients were newly diagnosed during the intervention and control periods, respectively (adjusted analysis: 26.7/1,000 versus 26.2/1,000, adjusted relative risk: 1.02 (95% confidence interval (CI) [0.94, 1.11]; $p = 0.634$)). The main limitation of this study is that it was conducted in a rapidly evolving epidemiological context.

## Conclusions

The results of this study showed that intensified screening for SARS-CoV-2 in EDs was unlikely to identify a higher proportion of newly diagnosed patients.

## Trial registration

**Trial registration number:** ClinicalTrials.gov NCT04756609.

---

Author summary

### Why was this study done?

- Undetected asymptomatic Severe Acute Respiratory Syndrome Coronavirus 2 (SARS-CoV-2) infections or paucisymptomatic infections with mild symptoms are responsible

for a substantial portion of transmissions, which has been a major challenge in managing the Coronavirus Disease 2019 (COVID-19) pandemic.

- Free and widely available screening has been offered in many countries to reach asymptomatic/paucisymptomatic infectious individuals and improve detection, reduce transmission, and help contain the pandemic.

- To our knowledge, the value of systematically offering screening during medical consultations, particularly in emergency departments (EDs), to identify asymptomatic/paucisymptomatic infectious individuals has not been evaluated.

### What did the researchers do and find?

- The objective of this trial was to evaluate whether an intensified screening strategy involving screening for asymptomatic/paucisymptomatic patients and routine screening of symptomatic and hospitalized patients (intervention) could lead to a higher proportion of new diagnoses compared with routine screening (control) in 18 EDs in the Paris metropolitan region.

- During the intervention period, 4,283 asymptomatic/paucisymptomatic patients were screened, leading to 224 new diagnoses. Overall, 1,859 patients were newly diagnosed during the intervention period versus 2,084 during the control period, corresponding, after analysis, to 26.7 new diagnoses/1,000 patients versus 26.2/1,000.

- The proportion of new diagnoses among asymptomatic/paucisymptomatic patients in EDs was higher than that in community screenings in the region. Comparison with findings from a mathematical model indicated that ED data allowed the estimation of regional incidences for asymptomatic/paucisymptomatic infections.

### What do these findings mean?

- The results of this study showed that the intensified screening strategy was unlikely to bring a substantial benefit to the detection of SARS-CoV-2 infections.

- The population screened in EDs appeared to be more affected by SARS-CoV-2 than the general population screened in the region. Intensified screening in EDs could allow access to individuals who are less likely to be reached by the general screening system.

- The main limitation of this study is that it was conducted in a changing epidemiological context, which encourages further exploration of screening strategies for asymptomatically transmitted respiratory viruses to better define the observatory role of EDs in emerging epidemics.

### Introduction

Countries have experienced several waves of the Severe Acute Respiratory Syndrome Coronavirus 2 (SARS-CoV-2) pandemic, with considerable health and economic impacts [1].

Containing the pandemic was challenged by undetected asymptomatic or paucisymptomatic infections, which are estimated to account for nearly half of all transmissions [2–4].

In many countries, along with nonpharmaceutical control measures, widely available and free screening for SARS-CoV-2 has been offered in communities to reach infected asymptomatic/paucisymptomatic individuals who would otherwise not be tested. Mass screening experiments have been conducted at the city or country level [5–7]. In Slovakia, a population-wide screening program in 2020 led to a reduction in the prevalence of infection [6]. In 2020 to 2021, community-based asymptomatic screening in Liverpool was associated with a reduction in Coronavirus Disease 2019 (COVID-19)-related hospital admissions compared with a control population in England [7]. Difficulties in reaching the most deprived populations have been noted in these screening programs [5,8], as well as technical constraints due to the high volume of tests [6] and/or the poor sensitivity of some antigenic tests. Overall, research evaluating the impact of SARS-CoV-2 screening programs on improving detection, reducing transmission, and contributing to containing the pandemic is scarce. A recent systematic review also noted that these studies were modeling studies or retrospective observational analyses not conducted in a controlled environment [9].

A possible wide screening strategy for identifying asymptomatic/paucisymptomatic individuals is to systematically offer screening during medical consultations, particularly in emergency departments (EDs). French EDs receive a volume equivalent to one-third of the French population each year [10,11] and provide access to a large portion of the population, including low-income groups who may be less reached by the general screening system. Therefore, intensified SARS-CoV-2 screening in EDs could contribute to epidemic control and provide an opportunity to observe viral evolution.

In addition to SARS-CoV-2 screening in EDs for symptomatic patients, screening at the time of hospital admission has been mandatory since mid-2020 in European countries to limit nosocomial transmission after studies showed that a high number of inpatients, up to 50% of new positive cases, were asymptomatic carriers [12,13]. Evaluations of screening practices in EDs for inpatients highlighted the value of the strategy during periods of high prevalence [13–17]. However, to our knowledge, no study has evaluated the benefit of extending screening to all patients who visit EDs in reaching asymptomatic and paucisymptomatic infectious individuals and helping improve detection. The lessons learned from the implementation and evaluation of such measures could be useful in preparing for future epidemics.

The objectives of this trial were to offer an intensified screening strategy in 18 EDs in the Paris metropolitan area before the broad vaccine roll-out in May 2021 to evaluate whether this strategy could contribute to a higher detection proportion of SARS-CoV-2 infections during a period covering the third wave caused by the SARS-CoV-2 Alpha variant.

## Methods

### Ethics approval

The protocol was approved by the Committee for Patient Protection Ile-de-France II (n˚ 20.12.21.34036 MS1 RIPH 2 HPS) and by the French Data Protection Authorities.

The study was also approved by the Scientific and Ethical Committee of Assistance Publique–Hopitaux de Paris (AP-HP) clinical data warehouse (IRB00011591) allowing to retrieve the ED flow data. AP-HP clinical data warehouse initiative ensures patients' information and consent regarding the approved studies through a transparency portal in accordance with European Regulation on data protection and authorization (number 1980120) from the National Freedom and Informatics Commission.

The trial was registered at ClinicalTrials.gov (NCT04756609).

## Study design and setting

A cluster-randomized, two-period, crossover trial was conducted from February 17, 2021 to May 31, 2021, to evaluate the value of an intensified SARS-CoV-2 screening strategy combining nurse-driven screening for asymptomatic/paucisymptomatic patients with routine screening practice (intervention) compared to routine screening practice alone (control) in 18 EDs. Routine screening practice included screening symptomatic or hospitalized patients. The protocol of the ANRS|MIE DEPIST-COVID trial is available in Appendix A in S1 File (ANRS-MIE refers to the "Agence nationale de recherche sur le sida et les hépatites virales | Maladies Infectieuses Emergentes").

The Paris metropolitan region, divided into 8 geographical departments, includes more than 12 million people [10]. Annual visits to the 18 EDs located in 5 geographical departments represent 28% of the ED visits in the region (Appendix B in S1 File). All 18 invited EDs agreed to participate.

## Selection of participants

Information about this study was provided to patients through posters in the EDs during each study period. An individual information note was given to patients who met the inclusion criteria (>18 years) on arrival to obtain their express informed consent to participate in this study and data collection during each study period, in accordance with French law (Appendix A in S1 File). During the intervention period, in addition to routine screening practice, all asymptomatic/paucisymptomatic patients were offered a questionnaire and a screening test.

## Randomization

The unit of randomization was the ED. The strategy (control/intervention) applied during the first period was assigned using a balanced block randomization process with stratification based on ED flow and, for feasibility concerns, whether the ED had a SARS-CoV-2 screening device (block size of 2 to 6). The alternative strategy was applied during the second period after a minimum 1-day washout interval. The allocation schedule was computer generated in SAS by an independent statistician.

The duration of each period was a minimum of 1 month with the possibility of a 15-day extension for the interested EDs, retaining similar durations for the same center in each period.

The patients and teams could not be masked to allocation due to the nature of the intervention.

## Control condition

During the control period, patients who were referred for suspected SARS-CoV-2 infection or who verbally reported symptoms of SARS-CoV-2 infection followed a specific patient pathway on admission that included physician-driven SARS-CoV-2 molecular screening. Patients who met the severity criteria (i.e., high acuity) [18] or who could be hospitalized were also screened. The practice was not modified in any way. For centers that had started with the intervention, the equipment provided was removed during the washout period, and EDs returned to their routine practice.

## Intervention

Before the intervention period, nurses participated in a 1-h training session, including an educational lecture and a test demonstration. The centers that did not have the equipment were provided with the machines (Appendix C in S1 File).

During the intervention period, in addition to routine screening for symptomatic patients or inpatients, a questionnaire was offered to the remaining asymptomatic/paucisymptomatic patients. The paper-based questionnaire on sociodemographic characteristics, SARS-CoV-2 symptoms and risk exposure was offered to patients who could provide consent (Appendix A in S1 File). The questionnaire was completed by the patient or with the help of the nurse. The questionnaire, further referred to as the DEPIST-COVID questionnaire, was not provided to patients with confirmed or highly suspected SARS-CoV-2 infection. The questionnaire was developed by the research team based on other questionnaires [19,20].

During triage assessment, a rapid screening test by nasopharyngeal swabs was offered by nurses on a 24-h basis to all patients who completed the questionnaire. The type of test was chosen according to the symptoms recorded on the questionnaire (asymptomatic versus paucisymptomatic).

Paucisymptomatic patients were defined as those with mild symptoms reported on the questionnaire that were not the reason for consultation or not spontaneously reported by the patient at registration.

For paucisymptomatic patients, samples were tested with the multiplex polymerase chain reaction (PCR) assay QIAstat-Dx respiratory SARS-CoV-2 panel, further referred to as the rapid multiplex respiratory virus test, which detects 21 viruses and bacteria and provides results within 70 min [21].

For asymptomatic patients, samples were tested with the Abbott ID NOW COVID-19 assay, an isothermal nucleic acid amplification assay based on a nicking and extension amplification reaction (NEAR) technique. The assay, further referred to as the rapid molecular SARS-CoV-2 test, targets the RNA-dependent RNA polymerase (RdRp) gene segment and provides results within 5 to 13 min [22].

Nasopharyngeal swabs were tested directly, without dilution in viral transport medium.

Patients with a positive SARS-CoV-2 test were cared for according to local procedures. During the intervention period, a clinical research assistant was on site 5 days a week to facilitate staff adherence and monitor data collection.

## Measurements

Data from the DEPIST-COVID questionnaires were entered into a database (CleanWEB Telemedicine Technologies SAS, Boulogne-Billancourt, France). ED flow data, including patient and center characteristics, and SARS-CoV-2 routine screening data were extracted from electronic ED databases. Community screening data routinely collected for all individuals screened in the Paris metropolitan region [23], including symptomatic status, were provided by Santé Publique France.

## Outcomes

The primary outcome was the proportion of newly diagnosed SARS-CoV-2–positive patients among all adults visiting the 18 EDs during the study period.

Patients who tested positive and whose SARS-CoV-2–positive status was not already known were considered positive. Unscreened patients or those whose test result was negative, indeterminate, or positive with a known status <3 months prior were classified as nonpositive.

The secondary outcomes were as follows: (a) implementation of ED screening for asymptomatic/paucisymptomatic patients, including the proportion of patients who completed the questionnaire; the proportion of screening tests that were offered, accepted, and performed; and patient characteristics reported in the DEPIST-COVID questionnaire; (b) the proportion of new SARS-CoV-2 infections diagnosed through ED screening for asymptomatic/

paucisymptomatic patients and patient characteristics, and the proportion of patients who tested positive for other respiratory viruses; (c) the proportions of new SARS-CoV-2 infections diagnosed through ED screening for asymptomatic/paucisymptomatic patients versus the proportions of positive tests through community screening for asymptomatic adults in the geographical departments of the Paris metropolitan region during the same period, overall and per ED; the proportion of new SARS-CoV-2 infections diagnosed through ED screening for all patients versus the proportion of positive tests through community screening for all adults in the geographical departments of the Paris metropolitan region during the same period; and (d) the proportions of new SARS-CoV-2 infections diagnosed through ED screening versus the incidence rate in the Paris metropolitan region estimated from an age-stratified disease model for SARS-CoV-2 transmission.

## Sample size calculation

The estimate of the sample size was based on the comparison of 2 proportions under individual randomization using Fisher's exact test, with 0.1% of newly diagnosed SARS-CoV-2–positive patients among the included patients (control) and 0.2% (intervention), with a 2-sided significance, $\alpha = 5\%$, and $\beta = 10\%$. Taking the intracluster (rho = 0.00014) and between-cluster (rho12 = 0.00009) correlations previously estimated [24] into account, 104 000 patients were needed.

The proportion of the control group was based on the incidence rate of 0.12% in the Paris metropolitan region on 26/11/2020. It was estimated that the proportion of 0.10% was close to the baseline proportion of new diagnoses among ED patients with routine screening. Simulations were performed using different incidence rates; the worst-case scenario involving the largest number of patients to be included was selected. The calculation was performed by an independent biostatistician with SAS software (version 9.4; SAS Institute, Cary, North Carolina, United States of America).

## Statistical analysis

Primary analysis was performed according to the intention-to-treat principle. The primary outcome was analyzed using a generalized linear mixed model (Poisson distribution) with the center as a random intercept, the center-by-period interaction as a random effect and the strategy (intervention versus control), period, and community incidence rate-by-period interaction as covariates.

The logarithm of the number of patients was included as an offset term. The parameters of the model were estimated using a full maximum-likelihood method with the Laplace method (SAS PROC GLIMMIX). The *P* values reported for covariates were based on *t* tests using the Kenward–Roger approximation for the denominator degrees of freedom. The period was modeled as a weekly categorical variable (1–15). The community incidence rate in the population of the local geographical department in the corresponding period and center was calculated among individuals aged 20+ years (data available only for this age category) [10,23]. Intercluster and intracluster correlation coefficients were calculated.

In sensitivity analyses, the period was modeled as a binary variable; variables used for randomization (screening equipment, flow) and intervention-by-week since roll-out interaction were included as covariates. Another sensitivity analysis was performed at the center level.

For patients with multiple tests in 24 h, a single result was considered; it was considered positive in the case of a positive result. ED visits and test results for the same patient separated by more than 24 h were considered separately.

The implementation of ED screening for asymptomatic/paucisymptomatic patients was described. Factors associated with performing ED screening for asymptomatic/paucisymptomatic patients were analyzed among patients who were offered screening using a generalized linear mixed model (binomial distribution) with the center as a random intercept and the patient characteristics as fixed effects. Patients who accepted screening but whose screening test was not performed for feasibility concerns were not included in the analysis. Missing data for covariates ranged from 0.2% to 15.6%.

The proportions of new SARS-CoV-2 infections and other respiratory virus infections diagnosed through ED screening for asymptomatic/paucisymptomatic patients were described, as well as patient characteristics. The factors associated with new SARS-CoV-2 diagnoses through this screening were analyzed using a generalized linear mixed model (binomial distribution) with the center as a random intercept and the patient characteristics as fixed effects. Missing data for covariates ranged from 0.6% to 12.8%.

In both models described above, the unit of analysis was the patient. Parameters were estimated with the Laplace method, and the fixed effects were estimated using the containment method. The variables that were found to be significant at $P < 0.05$ in bivariate analyses based on $t$ tests were included in the models. Observations with missing data were excluded.

The proportions of new SARS-CoV-2 infections diagnosed through ED screening for asymptomatic/paucisymptomatic patients overall and per ED, and the same proportion through ED screening for all patients were compared to the proportions of positive tests through community screening for adults screened in the geographical departments of the Paris metropolitan region using the Miettinen–Nurminen (MN) method [25–27] for risk difference and 95% confidence interval (CI) estimations. Pearson $X^2$ test was used to test for differences in these secondary outcomes.

Analysis for secondary outcomes used descriptive statistics, with numbers and proportions (qualitative data), and means, standard deviations or medians and the first quartile and third quartile (quantitative data). Differences and 95% CIs were estimated using the MN method (qualitative data) and the Brookmeyer–Crowley test (quantitative data).

The analysis plan for estimating the incidence rate in the Paris metropolitan region is detailed below. The proportion of new diagnoses was considered among ED-tested patients. The proportion was calculated for patients screened through ED screening for asymptomatic/paucisymptomatic patients and for all patients screened. The results were compared to the predictions of an age-stratified disease model for transmission in the region, previously validated with serological data and used in France in 2020 to 2021 [28,29]. The model integrated data on demographics, age profile, social contacts, mobility, and the adoption of preventive measures over time. It included data on the roll-out of the vaccination campaign and estimates of vaccine effectiveness and accounted for the cocirculation of the historical strain and the Alpha variant, which became dominant during the study period. The age classes were [0–11), [11–19), [19–65), and 65+ years. Transmission dynamics followed a compartmental scheme specific for COVID-19, where individuals were divided into susceptible/exposed/infectious/hospitalized/recovered stages. The infectious stage included a prodromic phase followed by a phase where individuals may continue to either be asymptomatic (probability: 40%) [4] or develop symptoms with degrees of severity (paucisymptomatic, mild/severe symptoms). Fitted to daily regional hospital admissions, the model allowed to estimate the total number of cases and the detection rate of virological monitoring in the region.

The analysis was performed using SAS software (version 9.4; SAS Institute, Cary, North Carolina, USA) and R 4.1.3 (2022-03-10). The code of the transmission model developed in Python 3 is available [28–30].

This study is reported as per the cluster randomized trials extension of the Consolidated Standards of Reporting Trials (CONSORT) guidelines (Appendix L in S1 File) [31].

## Results

Among the 73,340 patients who presented to the EDs during the intervention period and the 72,701 patients during the control period, 69,248 and 69,104 adults were included during the 2 periods, respectively, for a total of 138,352 patients (Fig 1). They had a median age of 45.0 years [31.0, 63.0], and women represented 45.7% of the patients. Patient characteristics are shown in Table 1.

### Intervention delivery

During the intervention period, the questionnaire was offered to 6,428 (13.6%) of 47,234 eligible patients (Fig 2). Among them, 6,332 (98.5%) patients agreed to complete the questionnaire. SARS-CoV-2 screening was offered to 5,984 (94.5%) asymptomatic/paucisymptomatic patients. Of the 1,793 patients offered screening for whom ED discharge data were available, 1,769 (98.7%) were not hospitalized (outpatients) (Appendixes D and K in S1 File).

Overall, 4,283 asymptomatic/paucisymptomatic patients were screened, including 3,682 (86.0%) patients who underwent a rapid molecular SARS-CoV-2 test, resulting in 224 (5.2%) new SARS-CoV-2 diagnoses. Among the asymptomatic/paucisymptomatic patients screened, 80.9% did not report symptoms and 41.6% reported not having been tested previously. Of the 224 newly diagnosed patients, 38.4% did not report symptoms (Appendix E in S1 File). Among the 601 patients who underwent a rapid multiplex respiratory virus test, 116 (19.3%) were newly diagnosed with SARS-CoV-2, including 5 diagnosed with other respiratory viruses, and 72 (12.0%) who tested negative for SARS-CoV-2 were positive for another virus.

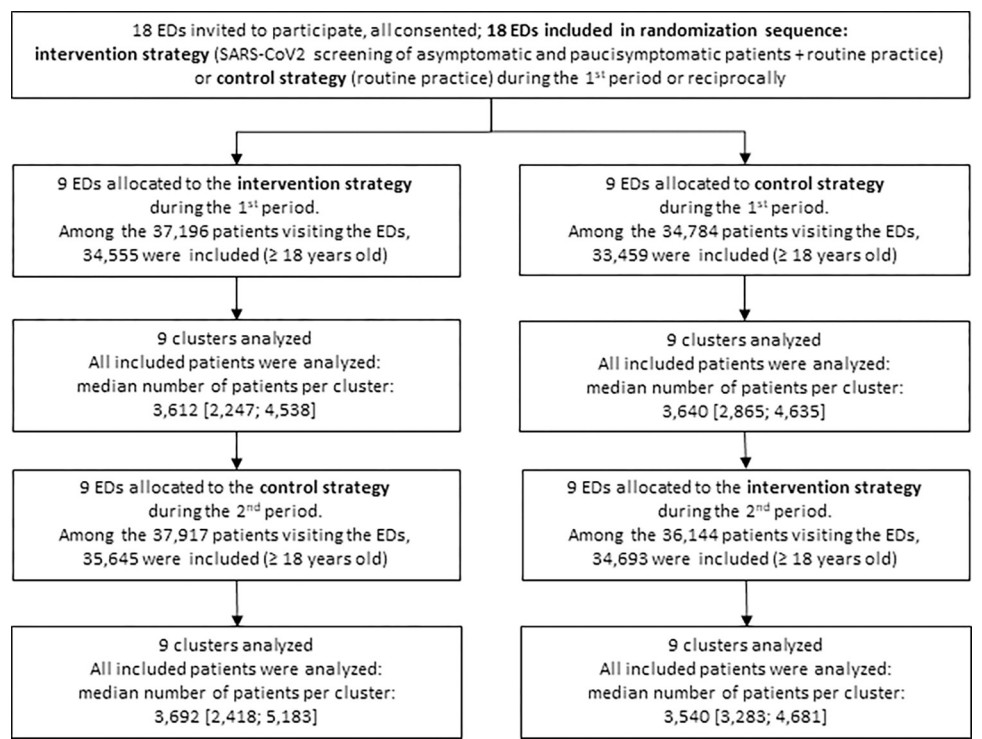

**Fig 1. Participation of emergency departments and patient inclusion in the DEPIST-COVID trial.** Data are presented as numbers or medians [first quartile; third quartile]. ED, emergency department.

**Table 1. Characteristics of the study participants.**

| Characteristic | Total population N = 138,352 | | Control period n = 69,104 | | Intervention period n = 69,248 | |
|---|---|---|---|---|---|---|
| | N | n (%) or median [Q1, Q3] | N | n (%) or median [Q1, Q3] | N | n (%) or median [Q1, Q3] |
| Age (year) | 132,865[1] | 45.0 [31.0, 63.0] | 66,237 | 45.0 [31.0, 64.0] | 66,628 | 45.0 [30.0, 63.0] |
| Sex | 127,868[2] | | 66,240 | | 61,628 | |
| Female | | 58,474 (45.7) | | 30,277 (45.7) | | 28,197 (45.8) |
| Center | 138,352 | | 69,104 | | 69,248 | |
| 1 | | 12,601 (9.1) | | 6,420 (9.3) | | 6,181 (8.9) |
| 2 | | 7,448 (5.4) | | 3,815 (5.5) | | 3,633 (5.2) |
| 3 | | 6,826 (4.9) | | 3,388 (4.9) | | 3,438 (5.0) |
| 4 | | 7,304 (5.3) | | 3,692 (5.3) | | 3,612 (5.2) |
| 5 | | 9,721 (7.0) | | 5,183 (7.5) | | 4,538 (6.6) |
| 6 | | 7,180 (5.2) | | 3,640 (5.3) | | 3,540 (5.1) |
| 7 | | 11,812 (8.5) | | 5,972 (8.6) | | 5,840 (8.4) |
| 8 | | 10,763 (7.8) | | 5,434 (7.9) | | 5,329 (7.7) |
| 9 | | 9,661 (7.0) | | 4,696 (6.8) | | 4,965 (7.2) |
| 10 | | 5,014 (3.6) | | 2,498 (3.6) | | 2,516 (3.6) |
| 11 | | 4,213 (3.0) | | 1,989 (2.9) | | 2,224 (3.2) |
| 12 | | 4,654 (3.4) | | 2,418 (3.5) | | 2,236 (3.2) |
| 13 | | 8,444 (6.1) | | 4,205 (6.1) | | 4,239 (6.1) |
| 14 | | 6,422 (4.6) | | 3,139 (4.5) | | 3,283 (4.7) |
| 15 | | 4,625 (3.3) | | 2,378 (3.4) | | 2,247 (3.2) |
| 16 | | 6,109 (4.4) | | 2,865 (4.1) | | 3,244 (4.7) |
| 17 | | 9,316 (6.7) | | 4,635 (6.7) | | 4,681 (6.8) |
| 18 | | 6,239 (4.5) | | 2,737 (4.0) | | 3,502 (5.1) |

[1] Missing data: n = 5,487 (4.0%). Age data were partially extracted at 2 centers.

[2] Missing data: n = 10,484 (7.6%). Sex data were partially extracted at 2 centers and were missing for 1 period at 1 center.

Q1, first quartile; Q3, third quartile.

During the intervention period, 17,512 patients were screened in routine practice, resulting in 1,635 (9.3%) new SARS-CoV-2 diagnoses.

In total, during this period, 1,859 patients were newly diagnosed among 69,248 patients (26.8/1,000 patients (95% CI [25.7, 28.1]) (Table 2).

## Control period

During the control period, following 18,572 tests, 2,084 patients (11.2%) were newly diagnosed among 69,104 patients (30.2/1,000 (95% CI [28.9, 31.5]) (Fig 3, Appendixes F–H in S1 File).

## Primary outcome

After adjustment, the proportion of new diagnoses was 26.7 per 1,000 patients (95% CI [19.1, 37.3]) during the intervention period versus 26.2 per 1,000 patients (95% CI [18.8, 36.3]) during the control period (adjusted relative risk: 1.02 (95% CI [0.94, 1.11]; $p = 0.634$)). In sensitivity analyses, the findings were similar. The intracluster and intercluster correlation coefficients are reported in Appendix I in S1 File.

## Secondary outcomes

The factors associated with performing screening for asymptomatic/paucisymptomatic patients were male sex (54.6% of patients screened versus 51.4% of patients who declined

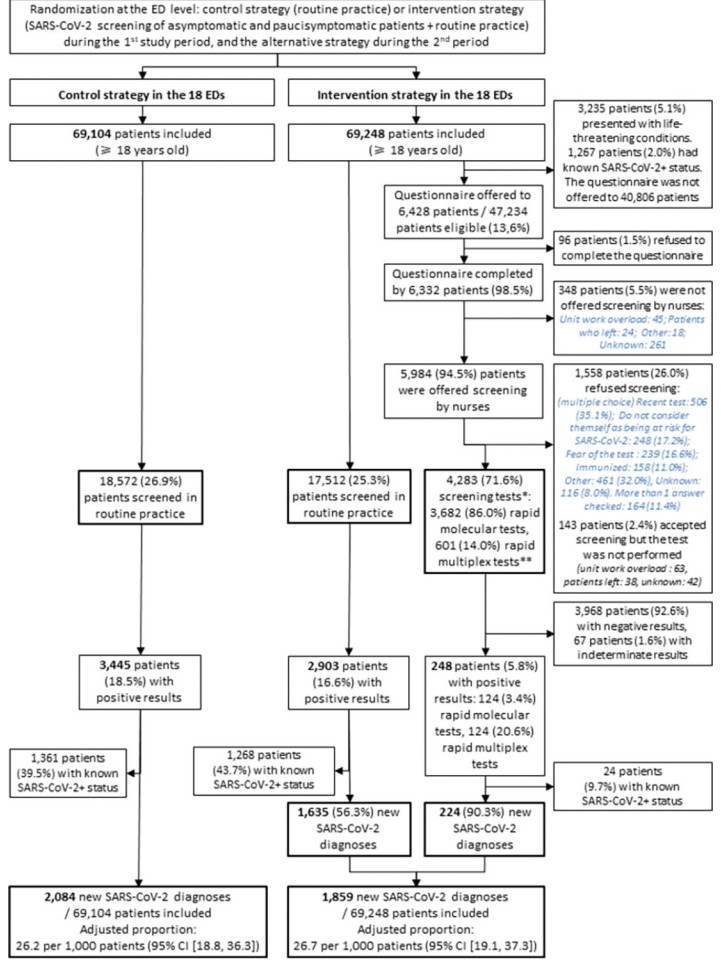

**Fig 2. Flow diagram.** For the primary outcome, all patients were analyzed according to the intention-to-treat principle. *Twenty-four patients with initial indeterminate SARS-CoV-2 results were retested: 2 had positive results, 16 had negative results, and 6 had indeterminate results. **The "rapid multiplex test" refers to the multiplex PCR assay QIAstat-Dx respiratory SARS-CoV-2 panel. The "rapid molecular test" refers to the Abbott ID NOW COVID-19 assay, an isothermal nucleic acid amplification assay based on a NEAR technique. CI, confidence interval; ED, emergency department; NEAR, nicking and extension amplification reaction; PCR, polymerase chain reaction; SARS-CoV-2, Severe Acute Respiratory Syndrome Coronavirus 2.

screening, OR: 1.21 (95% CI [1.05, 1.40])), mild symptoms (19.1% versus 6.4%, OR: 2.8 (95% CI [2.1, 3.7])), not being screened previously (41.6% versus 31.7%, OR: 1.4 (95% CI [1.2, 1.6])), and self-assessment of a high risk of infection (17.8% versus 14.2%, OR: 1.3 (95% CI [1.1, 1.6])) (Appendix D in S1 File).

**Table 2. Primary outcome data.**

| New SARS-CoV-2 diagnoses | Total population N = 138,352 | | Control period n = 69,104 | | Intervention period n = 69,248 | | Risk difference [95% CI] |
|---|---|---|---|---|---|---|---|
| | *N* | *n* (%) | *N* | *n* (%) | *N* | *n* (%) | |
| | 138,352 | | 69,104 | | 69,248 | | |
| Yes | | 3,943 (2.8) | | 2,084 (3.0) | | 1,859 (2.7) | −0.33 [−0.51, −0.16] |
| No | | 134,409 (97.2) | | 67,020 (97.0) | | 67,389 (97.3) | 0.33 [0.16, 0.51] |

CI, confidence interval; SARS-CoV-2, Severe Acute Respiratory Syndrome Coronavirus 2.

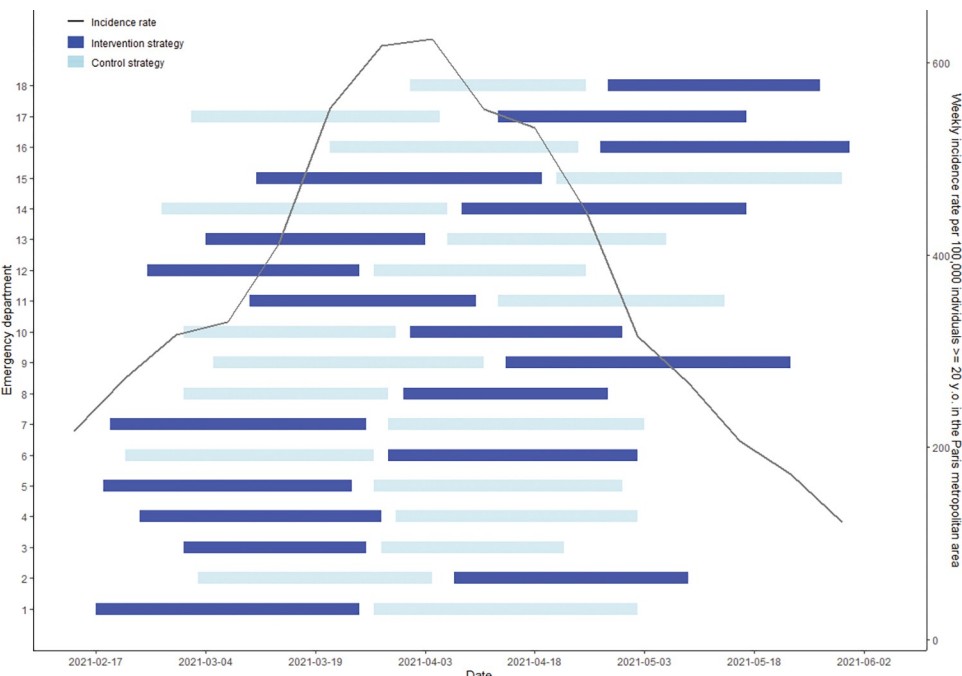

**Fig 3. Study periods by emergency department and weekly incidence rate.** Sources: Incidence: https://geodes.santepubliquefrance.fr/?view=map1&indics=sp_pe_std_quot.tx_std&serie=2022-01-31&lang=fr. Population: https://www.insee.fr/fr/statistiques/1893198. y.o., years old.

The factors associated with new diagnoses through screening for asymptomatic/paucisymptomatic patients were older age (OR: 1.02 (95% CI [1.01, 1.03]) per additional year), not being from France (54.4% of patients with new diagnoses versus 40% of patients who tested negative, OR: 1.7 (95% CI [1.2, 2.5])), particularly being from sub-Saharan Africa, having mild symptoms (61.6% versus 16.5%, OR: 7.3 (95% CI [5.1, 10.4])), being a case contact (29.7% versus 22.0%, OR: 1.8 (95% CI [1.2, 2.6])), and not being screened previously (57.7% versus 41.4%, OR: 2.1 (95% CI [1.5, 3.0])) (Appendix E in S1 File).

At the regional level, the proportion of new diagnoses through ED screening for asymptomatic/paucisymptomatic patients was higher than the proportion of positive tests through community screening for asymptomatic adults in the departments of the Paris metropolitan region (224/4,283 (5.2%) and 156,298/3,400,584 (4.6%), respectively, risk difference: 0.6% (95% CI [0.01%, 1.3%]; $p$ = 0.048)) (Appendix J in S1 File).

The proportion of new diagnoses through ED screening for all patients was higher than the proportion of positive tests through community screening for adults in the departments of the Paris metropolitan region (3,943/40,367 (9.8%) and 361,089/4,680,491 (7.7%), respectively, risk difference: 2.1% (95% CI [1.8%, 2.3%]; $p < 0.001$)) (Appendix J in S1 File).

The estimates from ED screening were compatible with the predictions of the mathematical model, yielding a regional incidence rate of asymptomatic/paucisymptomatic infections in adults of 4.1% (95% probability ranges [2.5%, 5.7%]) and a regional incidence rate in adults (symptomatic and asymptomatic) of 9.7% (95% probability ranges [6.1%, 13.8%]).

## Discussion

During this large-scale study in EDs, we found that an intensified screening strategy for SARS-CoV-2 was unlikely to identify a substantial proportion of new diagnoses. Compared to

the asymptomatic population screened in the region, the asymptomatic/paucisymptomatic ED population screened by nurses appears to have been more affected. Comparison with a mathematical model indicated that the ED data allowed for the estimation of regional incidences for asymptomatic/paucisymptomatic SARS-CoV-2 infections.

In this study, screening for asymptomatic/paucisymptomatic patients led to the detection of SARS-CoV-2 in more than 200 patients. Almost all these patients were outpatients who would not otherwise have been screened during their visit. Despite a trend in the primary outcome analysis toward an increased detection of infections with the intervention, the 95% CI provided no support for values of importance, ranging from an 11% increase, which is of clinical interest, to a 6% decrease. Several reasons may explain these findings.

First, 21% of the patients who visited the EDs were admitted and screened. When combined with the screening of symptomatic patients, more than 25% of patients were routinely screened. This high proportion of detection, facilitated by using fast molecular assays as point-of-care tests in half of the centers, is unprecedented and could have lowered the impact of the intervention.

Second, the findings could be related to a highly fluctuating evolution of the COVID-19 epidemic during the study [7]. During the control period, for a similar flow, more tests appeared to be ordered for symptomatic patients or inpatients than during the intervention period (3,445 versus 2,903), with a positivity proportion that appeared to be higher (18.5% versus 16.6%). The changing epidemiological context alongside the lockdown period (April 03, 2021 to May 03, 2021) was probably imperfectly captured in the analysis, which confirmed a weekly period effect and an interaction with the incidence rate. The lockdown must have led to changes with probably reduced outpatient numbers (Appendix K in S1 File). The reduction in ED visits during the wave may have limited the role of EDs as an observatory of the epidemic.

Third, out-of-hospital screening was generalized during the study period, possibly reducing the impact of the intervention. In 2021, 35 million tests were recorded in the Paris metropolitan area [23]. In 2023, as the pandemic state of emergency ends in most countries, community screening also abruptly decreases, which could renew the interest for ED observatories of the epidemics, as is the case in France for seasonal influenza.

In this study, almost half of the asymptomatic/paucisymptomatic patients screened had not been previously tested for SARS-CoV-2, suggesting limited access to health care. The proportion of new diagnoses in this asymptomatic/paucisymptomatic population in EDs seemed to be higher than that in the asymptomatic population screened during the same period in the region. Positivity appeared higher in EDs in individuals born in countries other than France, corroborating studies reporting higher infection rates in minority groups [32]. These results indicate that EDs may provide access to a large population, including more deprived patients who may be less reached by the general screening system. This is one of the main strengths of this study. Community-based SARS-CoV-2 screening programs struggled to reach a wide range of populations and observed lower participation in most disadvantaged neighborhoods [8]. When epidemic growth is substantial, high levels of repeat screening should be maintained [33,34]. The strategy in EDs can allow repeat screening and specifically include high-risk groups. The findings in community screening and EDs reiterate the importance of examining strategies that overcome the effects of social and health inequalities and may inform the development of future interventions in other epidemics.

Data from the implementation of the intensified SARS-CoV-2 screening strategy in EDs are difficult to compare. Most studies evaluating the effectiveness of SARS-CoV-2 screening in adult EDs either focus on screening symptomatic patients for diagnostic purposes or screening admitted patients to reduce nosocomial transmission, are modeling studies, or report on the accuracy of screening tests.

Evaluation of community-based SARS-CoV-2 screening programs for asymptomatic individuals have involved few controlled comparisons of key health outcomes, such as the detection of positive cases, timely isolation of infectious individuals and their close contacts, transmissions, hospital admissions, or deaths. Furthermore, the evaluations conducted have often failed to distinguish between the impact of mass screening and a combination of other control measures introduced simultaneously. Large-scale community screening evaluated in 3 countries at rapidly growing phases of the epidemic appeared to have the potential to reduce prevalence, transmission, or subsequent hospitalizations, at least in the short term [6,7,34]. However, given the heterogeneity of the study designs, objectives, outcomes, or the absence of a comparison group, the findings cannot be directly compared with those obtained in EDs. In a study conducted in Switzerland in 2021 with repeat screening of approximately 27,000 individuals over 8 weeks, the authors favorably concluded that the screening program led to a reduction in the incidence rate of up to 50%, with approximately 200 positive tests observed, which is a raw number similar to that obtained with ED screening for asymptomatic/paucisymptomatic individuals [34].

Comparative evaluation of screening activities is crucial for identifying effective and equitable responses [7]. The results of the present study raise the issue of the study design choice in a changing epidemiological landscape. The alternative of a cluster randomization and crossover could have been a before-and-after design, but this would have resulted in difficult-to-control biases and a lower level of evidence; therefore, the chosen design was appropriate.

There is concern that adding screening will have unintended consequences for ED workload. Contrasting results are reported on the impact of inpatient screening on ED length of stay [15,22]. Screening yield can also be limited by staff shortages, costs, or testing capacities, encouraging the adaptation of screening practices for asymptomatic inpatients and, similarly, for outpatients to individual-level risk assessment and regional incidence [13,16].

In terms of limitations, in addition to the factors described, including the high fluctuation during the epidemic which may have reduced the impact of the intervention, during this study, 14% of eligible ED patients completed the questionnaire, which was the first stage of the nurse-driven screening strategy for asymptomatic/paucisymptomatic individuals, 67% of whom were screened. This is inherent to screening strategies in EDs, in which the proportion of patients involved in the screening process rarely exceeds 20% [24,35]. However, considering the high volume of patients visiting EDs, the number of patients reached far exceeds that of other facilities in the same period.

A rapid isothermal amplification assay was used for biosafety reasons and rapid turnaround time [22]. Although the performances of such assays might be questionable compared to PCR techniques, several studies evaluating the same test we used reported high specificity (>97.5%) and high sensitivity (>98.5%) provided that tests were performed with dry swabs after nasopharyngeal sampling [36–39]. The test provided qualitative results, and therefore, residual and presymptomatic infections could not be distinguished. The aim was also to choose a test that would be easy and safe for nurses with no prior experience or qualifications in laboratory testing. The test had to provide quick results, similar to lateral flow assays, possess a higher sensitivity for identifying asymptomatic individuals and offer result traceability. Finally, rapid isothermal assays offer testing options that may be more affordable than single automated real-time PCR assays, particularly in terms of cost of the system.

For reasons of feasibility, our study could not include an evaluation of the costs in the 2 groups compared or an examination of the choice of tests. This is a major area since decisions about the adoption of control measures must reflect both effectiveness and costs [9].

Given the considerable asymptomatic transmission of SARS-CoV-2 infection, screening is a key nonpharmaceutical measure for identifying infectious individuals, limiting transmission,

and containing the epidemic. During a rapidly evolving epidemic phase, we observed that the intensified screening strategy in EDs was unlikely to increase detection. However, comparing ED screening data with a mathematical model was encouraging, as it allowed estimation of regional incidences for both asymptomatic/paucisymptomatic SARS-CoV-2 infections and total cases. This approach can potentially be used to estimate the size of the epidemic in the population and the underdetection of cases in community surveillance [28,40,41].

The findings underline the need to further explore, using robust evaluation methods, ED strategies for the detection of SARS-CoV-2, or asymptomatically transmitted respiratory viruses to better define the observatory role of EDs in emerging epidemics.

In conclusion, the results of this study showed that intensified ED screening was unlikely to bring a substantial benefit to the detection of SARS-CoV-2 infections. This strategy could facilitate access to a population that appears to be more affected than the population of the region and could help to estimate the incidence of asymptomatic/paucisymptomatic infections.

## Supporting information

**S1 File. Appendix A. Study protocol including the initial statistical analysis plan (SAP) as submitted to French data authorities, the final SAP, a summary of the changes in the SAP, and the DEPIST-COVID questionnaire. Appendix B. Figure Geographical location of the 18 emergency departments involved in the DEPIST-COVID trial (Paris metropolitan area).** Fig B1. Geographical location of the 18 emergency departments involved in the DEPIST-COVID trial (Paris metropolitan area). References: R Core Team (2022). R: A language and environment for statistical computing. R Foundation for Statistical Computing, Vienna, Austria. URL https://www.R-project.org/. Hijmans, R.J. (2022). raster: Geographic Data Analysis and Modeling. R package version 4.1.3. https://CRAN.R-project.org/package=raster. **Appendix C. Table Emergency department characteristics and study period duration.** Table C1. Emergency department characteristics and study period duration. Data are presented as numbers or No 0/Yes 1. ED: Emergency department. [a]Annual patient admissions included pediatric admissions. [b]Duration of the intervention, control, and wash-out periods (days), median [first quartile, third quartile]: 34.5 [30.0, 36.0]; 34.5 [30.0, 36.0]; 1.5 [1.0, 2.0]. **Appendix D. Table Characteristics of patients who underwent and declined SARS-CoV-2 screening for asymptomatic/paucisymptomatic patients during the intervention period.** Table D1. Characteristics of patients who underwent and declined SARS-CoV-2 screening for asymptomatic/paucisymptomatic patients during the intervention period. * Patients who accepted the rapid test but did not have a test performed were not included in this table. In univariate analysis, factors associated with performing screening for asymptomatic/paucisymptomatic patients were the following: older age, male sex, having mild symptoms, not being from France, being unemployed, being followed up for a chronic disease, not being screened previously, and self-assessment of a high risk of infection. ** Chronic diseases: diabetes, arterial hypertension, angina pectoris, chronic bronchitis, asthma or chronic respiratory disease. *** For this variable of the DEPIST-COVID questionnaire, data were collected in 8 centers, and it was not possible to collect them later in the remaining 10 centers. Findings are presented for 8 centers ($n$ = 2,226 patients, missing data: 19.5% in the 8 centers). Q1: first quartile; Q3: third quartile; CI: confidence interval. **Appendix E. Table Characteristics of patients tested through SARS-CoV-2 screening for asymptomatic/paucisymptomatic patients.** Table E1. Characteristics of patients tested through SARS-CoV-2 screening for asymptomatic/paucisymptomatic patients. * Patients with indeterminate results or a known positive SARS-CoV-2 status were not included in this table. In univariate analysis, factors associated with new infections diagnosed through nurse-driven screening for asymptomatic/

paucisymptomatic patients were mainly older age, having mild symptoms, not being from France, particularly being from sub-Saharan Africa, being unemployed, not having medical coverage, living in a community, being a case contact and not being previously screened. ** Among patients who underwent a rapid molecular SARS-CoV-2 test, 258 (7.4%) reported having symptoms. Among patients who underwent a rapid multiplex respiratory virus test, 508 (90.1%) reported having symptoms. *** Chronic diseases: diabetes, arterial hypertension, angina pectoris, chronic bronchitis, asthma, or chronic respiratory disease. **** For this variable of the DEPIST-COVID questionnaire, data were collected in 8 centers, and it was not possible to collect them later in the remaining 10 centers. Findings are presented for 8 centers ($n$ = 1,510 patients, missing data: 12.1% in the 8 centers). ***** Rhinovirus/Enterovirus: $n$ = 29 (37.7%), Coronavirus OC43: $n$ = 12 (15.6%), Coronavirus NL63: $n$ = 10 (13.0%), Human Metapneumovirus A+B: $n$ = 10 (13.0%), Respiratory Syncytial Virus A+B: $n$ = 6 (7.8%), Parainfluenza virus 3: $n$ = 3 (3.9%), Bocavirus: $n$ = 2 (2.6%), Parainfluenza virus 4: $n$ = 2 (2.6%), Bocavirus | Coronavirus HKU1: $n$ = 1 (1.3%), Coronavirus 229E: $n$ = 1 (1.3%), Influenza A: $n$ = 1 (1.3%). Among SARS-CoV-2+ patients: Rhinovirus/Enterovirus: $n$ = 2, Bocavirus: $n$ = 1, Parainfluenza virus 4: $n$ = 1, Respiratory Syncytial Virus A+B: $n$ = 1. Q1: first quartile; Q3: third quartile; CI: confidence interval. **Appendix F. Figure Newly diagnosed SARS-CoV-2-positive patients per emergency department and strategy.** Fig F1. Newly diagnosed SARS-CoV-2–positive patients per emergency department and strategy. **Appendix G. Figure SARS-CoV-2 incidence rate in the Paris metropolitan area and per geographical department (75, 77, 92, 93, 95) of the emergency departments involved in the study.** Fig G1. SARS-CoV-2 incidence rate in the Paris metropolitan area and per geographical department (75, 77, 92, 93, 95) of the emergency departments involved in the study. y.o.: years old. **Appendix H. Table Characteristics of patients newly diagnosed with SARS-CoV-2 infection.** Table H1. Characteristics of patients newly diagnosed with SARS-CoV-2 infection. Q1: first quartile; Q3: third quartile; CI: confidence interval. **Appendix I. Primary outcome modelling, sensitivity analyses, and intercluster and intracluster correlation coefficients.** Table I1. Coefficient estimates from the generalized linear mixed model with a Poisson distribution including the center as a random intercept, the center-by-weekly period interaction as a random effect and the strategy, weekly period and incidence rate-by-weekly period interaction as fixed effects (SAS PROC Glimmix). Table I2. Significant random effects. Table I3. Analysis considering a binarized period (0/1). Table I4. Analysis including variables used for randomization (screening equipment in emergency departments and flow in emergency departments). Table I5. Analysis including strategy * week_since_roll_out interaction. Table I6. Analysis at the center level. **Appendix J. Comparison of the proportions of new SARS-CoV-2 diagnoses through screening in emergency departments and of positive tests through community screening for individuals aged 18+ of the geographical departments of the Paris metropolitan area screened during the same period.** Table J1. New SARS-CoV-2 infections diagnosed through screening in emergency departments for asymptomatic/paucisymptomatic patients and positive tests through community screening for asymptomatic adults in the geographical departments of the Paris metropolitan area during the same period. * Source: Community screening data were provided by Santé Publique France on May 20, 2022. Table J2. New SARS-CoV-2 infections diagnosed through screening in emergency departments (EDs) for asymptomatic/paucisymptomatic patients per ED and positive tests through community screening for asymptomatic adults tested in the corresponding geographical department.* Source: Community screening data were provided by Santé Publique France on May 20, 2022. Table J3. New SARS-CoV-2 diagnoses through screening in emergency departments and positive tests through community screening for adults in the geographical departments of the Paris metropolitan area during the same period.* Source: Community

screening data were provided by Santé Publique France on May 20, 2022. **Appendix K. Figures Visits in the 18 emergency departments during the study period in 2021 and during the same period in 2019.** Fig K1. Visits to the 18 emergency departments during the same period of the year in 2019 and 2021. Source: Observatoire Régional des Soins Non Programmés (ORNSP). Activité des services d'urgences en Ile-de-France 2021 [Available from: https://orsnp-idf.fr/wp-content/uploads/2022/06/20220606_rapport_annuel_urgences_2021_VF.pdf. Published: June 2022. Accessed date: 07/06/2023]. Fig K2. Proportion of emergency department visits without hospital admission per week in the 18 emergency departments during the same period in 2019 and in 2021. Source: Observatoire Régional des Soins Non Programmés (ORNSP). Activité des services d'urgences en Ile-de-France 2021 [Available from: https://orsnp-idf.fr/wp-content/uploads/2022/06/20220606_rapport_annuel_urgences_2021_VF.pdf. Published: June 2022. Accessed date: 07/06/2023]. **Appendix L. Cluster randomised trials extension of the Consolidated Standards of Reporting Trials (CONSORT) checklist. Appendix M. Study group.**
(PDF)

## Acknowledgments

The authors would like to thank all participating patients, emergency department nurses, and emergency department teams, as well as the Clinical Research Unit Paris-East and the Clinical Research Center Paris-East teams for their support of the study, the Assistance Publique–Hopitaux de Paris (AP-HP) clinical data warehouse (EDS AP-HP) team, Sonia Larid and the "Observatoire Régional des Soins Non Programmés Ile-de-France" (ORSNP IDF) team, Nathalie de Castro, Renaud Piarroux, Martine Piarroux, Fatiha Djennaoui, Chloé McAvoy, and Mohamed Hamidouche, Santé publique France Île-de-France, France. This study has been labeled as a National Research Priority by the National Orientation Committee for Therapeutic Trials and other researches on Covid-19 (CAPNET). The investigators would like to acknowledge ANRS | Maladies infectieuses émergentes for their scientific support, and the French Ministry of Health and Prevention and the French Ministry of Higher Education, Research and Innovation for their support. For the purpose of Open Access, a CC-BY public copyright licence has been applied to the present document by the authors and will be applied to all subsequent versions up to the Author Accepted Manuscript arising from this submission.

## Author Contributions

**Conceptualization:** Judith Leblanc, Lisbeth Dusserre-Telmon, Anthony Chauvin, Tabassome Simon, Karla Hemming, Vittoria Colizza, Laurence Bérard, Arnaud Fontanet, Jérome LeGoff, Anne-Claude Crémieux.

**Data curation:** Judith Leblanc, Chiara E. Sabbatini.

**Formal analysis:** Judith Leblanc, Chiara E. Sabbatini, Karla Hemming, Vittoria Colizza.

**Funding acquisition:** Judith Leblanc, Jérome LeGoff, Anne-Claude Crémieux.

**Investigation:** Judith Leblanc, Lisbeth Dusserre-Telmon, Anthony Chauvin, Jérome Convert, Sonia Lazazga, Carole Jegou, Nabila Taibi, Sandrine Dautheville, Damien Zaghia, Camille Gerlier, Muriel Domergue, Florine Larrouturou, Florence Bonnet, Anne-Claude Crémieux.

**Methodology:** Judith Leblanc, Tabassome Simon, Karla Hemming, Vittoria Colizza, Laurence Bérard, Arnaud Fontanet, Jérome LeGoff, Anne-Claude Crémieux.

**Project administration:** Tabassome Simon, Sarah Salhi.

**Supervision:** Anthony Chauvin, Tabassome Simon, Laurence Bérard, Arnaud Fontanet, Sarah Salhi, Jérome LeGoff, Anne-Claude Crémieux.

**Writing – original draft:** Judith Leblanc, Tabassome Simon, Chiara E. Sabbatini, Karla Hemming, Vittoria Colizza, Jérome LeGoff, Anne-Claude Crémieux.

**Writing – review & editing:** Judith Leblanc, Lisbeth Dusserre-Telmon, Anthony Chauvin, Tabassome Simon, Chiara E. Sabbatini, Karla Hemming, Vittoria Colizza, Laurence Bérard, Jérome Convert, Sonia Lazazga, Carole Jegou, Nabila Taibi, Sandrine Dautheville, Damien Zaghia, Camille Gerlier, Muriel Domergue, Florine Larrouturou, Florence Bonnet, Arnaud Fontanet, Sarah Salhi, Jérome LeGoff, Anne-Claude Crémieux.

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
