## [Editor Report · Decision Letter 0]

6 Jun 2023

Dear Dr Leblanc, 

Thank you for submitting your manuscript entitled "INTENSIFIED SCREENING FOR SARS-CoV-2 IN 18 EMERGENCY DEPARTMENTS THE CLUSTER-RANDOMIZED, TWO-PERIOD, CROSSOVER DEPIST-COVID TRIAL" for consideration by PLOS Medicine.

Your manuscript has now been evaluated by the PLOS Medicine editorial staff and I am writing to let you know that we would like to send your submission out for external assessment.

However, we first need you to complete your submission by providing the metadata that are required for full assessment. To this end, please login to Editorial Manager where you will find the paper in the 'Submissions Needing Revisions' folder on your homepage. Please click 'Revise Submission' from the Action Links and complete all additional questions in the submission questionnaire.

Please re-submit your manuscript within two working days, i.e. by Jun 08 2023 11:59PM.

Once your full submission is complete, your paper will undergo a series of checks in preparation for external assessment. 

Kind regards,

Richard Turner PhD

Consulting Editor, PLOS Medicine

plosmedicine@plos.org

---

## [Decision Letter · Decision Letter 1]

17 Aug 2023

Dear Dr. Leblanc,

Thank you very much for submitting your manuscript "INTENSIFIED SCREENING FOR SARS-CoV-2 IN 18 EMERGENCY DEPARTMENTS THE CLUSTER-RANDOMIZED, TWO-PERIOD, CROSSOVER DEPIST-COVID TRIAL" (PMEDICINE-D-23-01467R1) for consideration at PLOS Medicine. 

Your paper was evaluated by an associate editor and discussed among all the editors here. It was also discussed with an academic editor with relevant expertise, and sent to independent reviewers, including a statistical reviewer. The reviews are appended at the bottom of this email and any accompanying reviewer attachments can be seen via the link below:

[LINK]

In light of these reviews, I am afraid that we will not be able to accept the manuscript for publication in the journal in its current form, but we would like to consider a revised version that addresses the reviewers' and editors' comments. Obviously we cannot make any decision about publication until we have seen the revised manuscript and your response, and we plan to seek re-review by one or more of the reviewers. 

We expect to receive your revised manuscript by Sep 06 2023 11:59PM. Please email us (plosmedicine@plos.org) if you have any questions or concerns.

We look forward to receiving your revised manuscript. 

Sincerely,

Alexandra Schaefer, PhD

PLOS Medicine

plosmedicine.org

GENERAL COMMENTS

Please respond to all editor and reviewer comments.

Please cite the reference numbers in square brackets (e.g., “We used the techniques developed by our colleagues [19] to analyze the data”). Citations should be preceding punctuation.

Please cite your Supporting Information as outlined here: https://journals.plos.org/plosmedicine/s/supporting-information

Causal language - In trials, there is usually a distinction in the language in terms of causal vs associational for primary and secondary trial outcomes. It would be beneficial to use associational language in the discussion and other sections for secondary outcomes.

ACADEMIC EDITOR COMMENTS

- I agree with one of the reviewers who said that there should be more information on the screening of the community in Paris. How was this data collected and how representative is the screened population?

- It was at some points unclear whether authors were referring to the data from community screening or to results from modelling. Maybe clarify upfront that these two sources are used for comparison with the general population and clarify throughout the manuscript which of two you are referring to.

- In the tables, the term "control strategy" for denoting data collected during control periods could potentially be misleading. In my understanding, the term "control strategy" refers to a strategy to control the pandemic, and not control in the sense of control group. I suggest to write "control periods" and "intervention periods" or something similar.

COMPETING INTEREST STATEMENT

All authors must declare their relevant competing interests per the PLOS policy, which can be seen here: https://journals.plos.org/plosmedicine/s/competing-interests

For authors with ties to industry, please indicate whether any of the interests has a financial stake in the results of the current study.

FINANCIAL DISCLOSURE

The funding statement should include: specific grant numbers, initials of authors who received each award, URLs to sponsors’ websites. Also, please state whether any sponsors or funders (other than the named authors) played any role in study design, data collection and analysis, the decision to publish, or preparation of the manuscript. If they had no role in the research, include this sentence: “The funders had no role in study design, data collection and analysis, decision to publish, or preparation of the manuscript.”

DATA AVAILABILITY STATEMENT

PLOS Medicine requires that the de-identified data underlying the specific results in a published article be made available, without restrictions on access, in a public repository or as Supporting Information at the time of article publication, provided it is legal and ethical to do so. Please see the policy at http://journals.plos.org/plosmedicine/s/data-availability and FAQs at http://journals.plos.org/plosmedicine/s/data-availability#loc-faqs-for-data-policy 

The Data Availability Statement (DAS) requires revision. For each data source used in your study: 

In accordance with ICMJE requirements, PLOS Medicine requires prospective, public registration of a data sharing plan (as part of mandatory clinical trials registration) for all clinical trials that began enrollment on or after January 1, 2019.

ABSTRACT

Please report your abstract according to CONSORT for abstracts, following the PLOS Medicine abstract structure (Background, Methods and Findings, Conclusions); https://www.equator-network.org/reporting-guidelines/consort-abstracts/

Please ensure that all numbers presented in the abstract are present and identical to numbers presented in the main manuscript text.

PLOS Medicine requests that main results are quantified with 95% CIs as well as p values. Please include. When reporting p values please report as p<0.001 and where higher as the exact p value p=0.002, for example. For the purposes of transparent data reporting, if not including the aforementioned please clearly state the reasons why not.

Please include any important dependent variables that are adjusted for in the analyses.

Throughout, suggest reporting statistical information as follows to improve clarity for the reader “22% (95% CI [13%,28%]; p</=)”. Please amend throughout the abstract and main manuscript.

Please note the use of commas to separate upper and lower bounds, as opposed to hyphens as these can be confused with reporting of negative values.

When a p value is given, please specify the statistical test used to determine it. Please report p values as p<0.001 and where higher as 'p=0.002'

Please state that analysis was intention to treat.

Please provide the number of participants lost to follow up in each group.

Abstract Background: Provide the context of why the study is important. The final sentence should clearly state the study question.

Abstract Methods and Findings:

*Please provide brief demographic details of the study population (e.g. sex, age). Please define the time frame during which the study took place.

* Please include a summary of adverse events if these were assessed in the study.

Abstract Conclusions:

Please begin your Abstract Conclusions with "In this study, we observed ..." or similar, to summarize the main findings from your study, without overstating your conclusions. Please emphasize what is new and address the implications of your study, being careful to avoid assertions of primacy.

l.40: Please define ‘aRR’.

l.41: Please define ‘CI’.

AUTHOR SUMMARY

At this stage, we ask that you include a short, non-technical Author Summary of your research to make findings accessible to a wide audience that includes both scientists and non-scientists. The Author Summary should immediately follow the Abstract in your revised manuscript. This text is subject to editorial change and should be distinct from the scientific abstract. Please see our author guidelines for more information: https://journals.plos.org/plosmedicine/s/revising-your-manuscript#loc-author-summary.

The summary should include 2-3 single sentence, individual bullet points under each of the questions. The last bullet point should describe the main limitation(s) of the study's methodology.

It may be helpful to review currently published articles for examples which can be found on our website here https://journals.plos.org/plosmedicine/

INTRODUCTION

Please address past research and explain the need for and potential importance of your study. Indicate whether your study is novel and how you determined that. If there has been a systematic review of the evidence related to your study (or you have conducted one), please refer to and reference that review and indicate whether it supports the need for your study. 

l.65: Please revise tense. We suggest ‘The objectives of this study were to…’ or similar.

METHODS AND RESULTS

Thank you for providing the completed CONSORT extension to cluster randomised trials checklist. Please ensure that all components of CONSORT are present in the manuscript, including how randomization was performed, allocation concealment, blinding of intervention, definition of lost to follow-up, power statement. Please include the completed CONSORT checklist as Supporting Information. When completing the checklist, please use section and paragraph numbers, rather than page numbers. 

Please add the following statement, or similar, to the Methods: "This study is reported as per the Consolidated Standards of Reporting Trials (CONSORT) guideline for cluster randomised trials (S1 Checklist)."

If the study included dropouts, specify whether their data are imputed and if so using what method, please refer to as modified ITT. 

PLOS Medicine requests that main results are quantified with 95% CIs as well as p values. When reporting p values please report as p<0.001 and where higher as the exact p value p=0.002, for example. For the purposes of transparent data reporting, if not including the aforementioned please clearly state the reasons why not.

Please include any important dependent variables that are adjusted for in the analyses.

Suggest reporting statistical information as detailed above – see under ABSTRACT

Please present numerators and denominators for percentages, at least in the Tables [not necessarily each time they're mentioned].

Where questionnaires were used, please provide a copy of the questionnaire in the supplementary files.

Please define "lost to follow-up" as used in this study. Other reasons for exclusion should be defined.

Please define the length of follow up (eg, in mean, SD, and range).

Please provide the actual numbers of events for the outcomes, not just summary statistics or ORs.

Please specify whether informed consent was written or oral.

The following secondary outcomes measures “For symptomatic patients: description of the symptomatology”, “Number of close contact persons who have been tested for SARS-Cov2, who have been tested positive and who have been isolated over both periods.”, “Sociodemographic, behavioural factors and habits associated with SARS-CoV-2 infection in patients tested in the following three schemes: DEPIST-COVID (positive cases, negative controls), ComCor (cases,

controls) and COVISAN” and “Estimate of incidence in the region, estimate of underdetection of cases in the region, and comparison with results from the model experiment” appear to differ between the submitted manuscript and the trial registry/study protocol. Please clarify and explain the discrepancy. (a) Can you please present those results as part of this manuscript, or indicate why that is not possible? (b) Can you please indicate when you plan to publish those results?

Please include the study protocol document and analysis plan, with any amendments, as Supporting Information to be published with the manuscript if accepted.

l.71: Please define ‘ANRS’.

l.94: “duration of each period was a minimum of 1 month with the possibility of a 15-day extension”. In the study procotol it was describe that duration of inclusion was intended to be 2 months (with the possibility of extending each period by a maximum of 15 days; 2 months and 30 days maximum + wash-out?. Please clarify and explain the discrepancy.

l.121: Please define ‘PCR’.

ll.142-157: Please improve the presentation of the secondary outcomes with special attention to clarity and structure.

l.189: “Implementation measures were described”. I am not sure what this sentence is intended to mean. Please revise this sentence.

ll.195-196: “- The proportion of new SARS-CoV-2 and other respiratory virus infections diagnosed through screening for asymptomatic/paucisymptomatic patients and patient characteristics were described.”. Please revise this sentence.

l.213: “- Incidence rate estimation in the Paris metropolitan region” – is this supposed to be a subheader? Please revise.

ll.189-213: Please remove the bullet points from your text and revise the paragraphs. 

ll. 231-233: Please remove the “Role of the funding source” paragraph. This statement should only be included in the according section in the online submission form.

DISCUSSION

Please present and organize the Discussion as follows: a short, clear summary of the article's findings; what the study adds to existing research and where and why the results may differ from previous research; strengths and limitations of the study; implications and next steps for research, clinical practice, and/or public policy; one-paragraph conclusion. Please remove any subheadings.

l.331: We have noted that this is the first time you have cited Supplementary Figure S11. Please note that all figures and tables (including those in Supporting information files) should be mentioned in the main text before the discussion.

FIGURES

For all Figures, please ensure that you have complied with our figures requirements http://journals.plos.org/plosmedicine/s/figures.

Please provide titles and legends for all figures (including those in Supporting Information files).

Please consider avoiding the use of red and green in order to make your figure more accessible to those with colour blindness.

In the flow diagram, please indicate the number of individuals in each group analyzed in the ITT analysis. 

Please in the figure legend/description, define abbreviations used in each figure (including those in Supporting Information files).

Figure 1: Please check if '1st period' and '2nd period' need to be swapped in the right arm of the flowchart. As it stands, all Emergency Departments would have followed the intervention strategy during the 1st period of the trial.

Figure 1: Please change the ‘(18 years old or older)’ to ‘(≥ 18 years old)’.

Figure 2: Please define the ‘rapid molecular tests’ and ‘rapid multiplex respiratory virus tests’ in the figure legend. Are the ‘multiplex tests” (248 patients box) the same as the “rapid multiplex respiratory virus tests”? Please use the same descriptions to avoid confusion.

Figure 3: The figure title ‘Calendar’ is a very vague description of the figure. Please revise. 

TABLES

Please note the use of commas to separate upper and lower bounds, as opposed to hyphens as these can be confused with reporting of negative values. Suggest reporting statistical information as detailed above – see under ABSTRACT

Please provide titles and legends for all tables (including those in Supporting Information files).

Please define all abbreviations used in the table below each table (including those in Supporting Information files).

Table 1: Please define ‘med‘ and ‘Q‘.

Table 2: Please define ‘CI’.

SUPPLEMENT

eSupplement S3: The footnote ‘b’ is not in the actual table (mistakenly marked as ‘a’ following ‘(days)’). Please revise. In addition, please define ‘IQR’.

eSupplement S4: Under ‘Medical Coverage’, please change ‘or Autres’ to ‘or other’. Please define ‘CI’, ‘med’, ‘Q’.

eSupplement S5: Under ‘Medical Coverage’, please change ‘or Autres’ to ‘or other’. Please define ‘CI’, ‘med’, ‘Q’, ‘ED’.

eSupplement S5: For ‘Fever (>38°) or feeling Feverish’, please add a unit (Celsius).

eSupplement S6: Please define ‘ED’ or change to ‘Emergency department’.

eSupplement S7: Please define ‘y.o.’. 

eSupplement S8: Please define ‘med’, ‘Q’, ‘CI’.

eSupplement S9/eTable 1: Please define ‘CI’.

eSupplement S9/eTable 2: Please define ‘SE’.

eSupplement S9/eTable 3/5/6: Please define ‘SE’, ‘CI’.

eSupplement S9/eTable 4: Please define ‘ED’, ‘SE’, ‘CI’.

eSupplement S10: Please define ‘ED’, ‘CI’.

eSupplement S11: Please define ‘ED’. Please increase the size of the figure legend and the axis label. Please revise throughout the entire manuscript.

REFERENCES

PLOS uses the numbered citation (citation-sequence) method and first six authors, et al.

Please ensure that journal name abbreviations match those found in the National Center for Biotechnology Information (NCBI) databases (http://www.ncbi.nlm.nih.gov/nlmcatalog/journals), and are appropriately formatted and capitalised.

Please also see https://journals.plos.org/plosmedicine/s/submission-guidelines#loc-references for further details on reference formatting. 

Where website addresses are cited, please specify the date of access. 

Comments from the reviewers:

Reviewer #1: The Authors present an very interesting manuscript about the value of screening for asymptomatic or paucisymptomatic patients in addition to routine screening for the detection of SARSCoV-2 infection in the emergency departments (EDs). The intensified screening strategy took place in the year 2021 in EDs of the Paris metropolitan area, in addition to routine screening practice. The intensified screening strategy consisted of a questionnaire about risk exposure and symptoms offered by nurses to asymptomatic/paucisymptomatic patients from February 17, 2021, to May 31, 2021. The primary outcome was the proportion of newly diagnosed SARS-CoV-2-positive patients among all adults visiting EDs. 

During the intervention period, 17,512 patients were screened in routine practice, resulting in 1,635 (9.3%) new SARS-CoV-2 diagnoses while 4,283 patients were screened with the intensified screening strategy , resulting in 224 new diagnoses (5.2%). In total, 1,859 patients were newly diagnosed among 69,248 ED admitted patients. The Authors conclusion is that an intensified ED screening strategy is unlikely to identify a substantial proportion of new diagnoses if the intensified screening is aimed at asymptomatic patietns who are unlikely to refer to ED. This data is functional to the concern that adding SARS CoV-2 screening in ED will have unintended consequences for ED workload, work capacity and length of stay.

I would ask the Authours a brief comment in the discussion on the cost of using the ID NOW molecular platform in ED setting in their designed strategy of intensifying SARS CoV-2 testing.

Reviewer #2: This manuscript by Leblanc et al. evaluates an expanded and targeted SARS-CoV-2 screening strategy intervention in 18 EDs in the Paris region using a cluster-randomized, two-period, crossover trial serendipitously performed over a major peak of regional SARS-CoV-2 transmission between February and June of 2021. The authors demonstrate that their expanded screening strategy did not identify a significantly higher proportion of new cases compared to the control screening methodology during this dynamic interval of rapidly changing disease prevalence. The trial was expertly performed, the analyses are thorough and robust, and the manuscript is well-written. I offer the following suggestions to improve the breadth of interpretations in the discussion.

1) In response to a previous round of reviews, the authors have appropriately tempered initial claims of systematic screening and used the terminology "intensified screening." 

The authors' study seems most consistent with a "targeted" screening strategy as it relies upon nurse-directed implementation of a questionnaire prior to selecting an appropriate screening test for the disease of interest. As others have previously shown in a low prevalence disease setting (doi: 10.1001/jamanetworkopen.2021.17763), targeted screening strategies do not typically increase the number of new diagnoses when the prevalence of disease is low but do reduce the number of tests administered to the population of interest. The present study was performed during a period of very high disease prevalence and an untargeted study design (free testing offered to anyone without the implementation of an initial questionnaire) may have achieved much higher than 13.6% participation in the screening process which may have corresponded to more cases diagnosed. While previously published data evaluating low-prevalence disease settings do not suggest that this would have changed the authors results or conclusions, it is important to note in the discussion that the 13.6% participation rate was potentially lower than it otherwise might have been due to the targeted nature of the screening process.

It is also unclear in the setting of a very high prevalence disease setting if increased participation rates might have changed the results in favor of universal screening - emphasizing the importance of being very clear that the present study evaluated a targeted screening strategy rather than an untargeted strategy. Clearly, the authors had no way to know during study planning that the study interval would be perfectly timed to coincide with the peak of a major outbreak of disease, therefore their design of a targeted screening strategy was completely justified, and this is not a major study limitation - just one that should be made as transparent as possible in the discussion.

The authors should address in the discussion their results that further support that their screening methodology was ultimately targeted - especially the fact that the factor most strongly associated with performing screening for asymptomatic/paucisymptomatic patients was the presence of mild symptoms (OR 2.1-3.7, line 290). The higher frequency of disease found in the 4,283 screening tests offered during the intervention period when compared with the 3.4 million screening tests performed in the Paris Metropolitan region in the same time interval also strongly suggests that there was some selection bias in who received the testing. Whether this was a location-specific bias because all these patients were seen in an acute care medical setting or if this was related to the selection bias of the specific patients who chose to take / were offered the questionnaire cannot be deduced from the present data. Nevertheless, being clear in the discussion that these results together with the study design suggest that the screened population was targeted will allow the practicing ED clinician to realize that the present results are only generalizable to a targeted screening strategy implemented during a high-prevalence interval during a pandemic/epidemic wave rather than to an untargeted screening strategy implemented at those same times or to a targeted screening strategy implemented at a time of low disease prevalence.

2) While not a limitation of the authors present study, the authors should consider adding a paragraph to the discussion of the consequences of implementing a test such as the rapid isothermal amplification assay with a specificity of > 97.5% during a period of low disease prevalence. They appropriately address this last comment from the previous reviewer #1 by stating that it is not an issue in their study where there was such high prevalence of disease. However, those who are contemplating expanding asymptomatic screening during periods of low disease prevalence, as that reviewer appeared to be advocating, will need to consider tests with only the highest specificity or the proportion of false positive results can rapidly diminish the utility of any screening strategy. The authors should comment how any future considerations of expanded screening for viral respiratory diseases should consider both the prevalence of disease and the specificity and sensitivity of individual tests in addition to cost-effectiveness. This has become acutely apparent in the past several months as some hospital systems continue to insist upon universal SARS-CoV-2 screening strategies for admitted patients when the prevalence of disease has diminished substantially.

Reviewer #3: Statistical review

This paper reports a cluster-crossover trial evaluating a screening strategy in emergency departments for covid. It is shown that this doesn't significantly change the number of infections identified. The trial is analysed with appropriate methods and is reported very well. I have some minor comments:

1. Abstract: In the findings paragraph it would be useful to add more about the number screened in the Paris metropolitan region.

2. Methods: I also found information lacking about the screening in Paris metropolitan region - was this routinely collected data or conducted as part of the research study? It's possible this estimate was treated as a known, fixed value in which case I would recommend this is made clearer in the paper.

3. Statistical analysis: where 'patient characteristics' are referred to, I would recommend these are provided somewhere.

4. Page 7: "estimated using the KR or containment method." - what affected which method was used?

5. Page 8: The "Miettinen-Nurminen (MN)" method was new to me, can more be said about what this does and justifying its use? Is this because the data available was a 2x2 table?

6. Page 8: "calculated by profile (diagnoses through screening for asymptomatic…". I would recommend clarifying this sentence: "The proportion was calculated separately for different profiles of patients. These profiles included 'diagnoses through screening for asymptomatic/paucisymptomatic patients' and 'all cases'." If something else is meant by calculated by profile, I would make this clearer.

7. Page 11: I think 'IC on line 300' should be CI.

James Wason

[LINK]

---

## [Decision Letter · Decision Letter 2]

18 Oct 2023

Dear Dr. Leblanc,

Thank you very much for re-submitting your manuscript "INTENSIFIED SCREENING FOR SARS-CoV-2 IN 18 EMERGENCY DEPARTMENTS THE CLUSTER-RANDOMIZED, TWO-PERIOD, CROSSOVER DEPIST-COVID TRIAL" (PMEDICINE-D-23-01467R2) for review by PLOS Medicine.

I would like to thank you for your considered and detailed responses to the editors' and reviewers' comments. I have discussed the paper with my colleagues and the academic editor, and it has also been seen again by two of the original reviewers. The changes made to the paper were satisfactory to the reviewers, and any remaining comments are editorial in nature. As such, we intend to publish the paper in PLOS Medicine. However, before we can officially accept the paper, we kindly request that you address a few additional editorial comments (found below). When submitting your revised paper, please include once again a detailed point-by-point response to the editorial comments.

[LINK]

We expect to receive your revised manuscript within 1 week. Please email me (aschaefer@plos.org) if you have any questions or concerns.

If you have any questions in the meantime, please contact me (aschaefer@plos.org) or the journal staff on plosmedicine@plos.org.  

We look forward to receiving the revised manuscript by Oct 25 2023 11:59PM.   

Sincerely,

Alexandra Schaefer, PhD

Associate Editor 

PLOS Medicine

plosmedicine.org

Requests from Editors:

TITLE

1) We suggest changing the title to: “Intensified screening for SARS-CoV-2 in emergency departments in Paris, France (DEPIST-COVID): A cluster-randomized, two-period, crossover trial”

FINANCIAL DISCLOSURE

1) If possible, please include specific grant numbers in your funding statement.

DATA AVAILABILITY STATEMENT

1) Thank you for updating your Data availability statement. We suggest changing the statement as follows: “The study data are owned by the Assistance Publique - Hôpitaux de Paris (AP-HP) sponsor, “Département de la Recherche Clinique et du Développement” and are not freely available. Requests to access the data should be directed to: DJENNAOUI Fatiha <fatiha.djennaoui@aphp.fr> and [DRC] Secretariat Promotion Délégation à la Recherche Clinique et à l'Innovation <drc-secretariat-promotion@aphp.fr>.”

ABSTRACT

1) We noted that you provided the CONSORT Abstract checklist as a supplementary file. You do not need to provide the actual file, but we would like to encourage you to revise your abstract once more following the CONSORT Abstract checklist. We have pointed out some specific items in the following points.

2) Please only report the primary outcome of the trial in your abstract. We are in the process of overhauling our workflows and policies (with a new Executive Editor), and we now require that trial abstracts only report secondary outcomes if all secondary outcomes of the trial are included. For trials that have many secondary outcomes, such as yours, the abstract should be limited to reporting the primary outcome.

3) At the beginning of the Methods/Findings section of the abstract, please change ‘During this cluster-randomized…’ to ‘We conducted a cluster-randomized…’.

4) Please include the clinical trial registry number in the abstract.

5) In the last sentence of the Abstract Methods and Findings section, please describe the main limitation(s) of the study's methodology (e.g., “The main limitation of the study is that it was conducted in a rapidly evolving epidemiologic context.”)

6) “The ED population seems to be more affected than the population of the region” – this statement and the conclusion paragraph seem rather unclear. Please revise the conclusion paragraph for clarity and aim to focus on the study implications and the interpretation of the study based on the primary results presented in the abstract.

AUTHOR SUMMARY

1) The Author Summary requires revision. Under the second question ‘What Did the Researchers Do and Find?, the bullet points, particularly the second one, are rather long and we encourage you to focus on the most important outcome (i.e. the primary outcome). To improve clarity, we also suggest changing the bullet points under ‘Why Was This Study Done?’ as follows:

• Undetected asymptomatic SARS-CoV-2 infections or paucisymptomatic infections with mild symptoms are responsible for a substantial portion of transmissions, which has been a major challenge for managing the COVID-19 pandemic. 

• Free and widely available screening has been offered in many countries to reach asymptomatic or paucisymptomatic infectious individuals and improve detection, reduce transmission, and help contain the pandemic.

• The value of systematically offering screening during medical consultations, particularly in emergency departments (EDs), as a possible strategy to identify asymptomatic/paucisymptomatic infectious individuals has, to our knowledge, not been evaluated.

2) The last bullet under ‘What Do These Findings Mean?’ point should describe the main limitation of the study's methodology.

INTRODUCTION

1) ll.119-120, suggest: A recent systematic review also pointed out these studies were modeling studies or retrospective observational analyses not conducted in a controlled environment [9].

2) l.137: Please add the time point for the “vaccine roll-out”.

METHODS AND RESULTS

1) ll.317-320: There seems to be a formatting issue – please revise.

FIGURES

1) Figure 3: Please include in the figure description an explanation of the meaning of the black line.

CHECKLIST 

1) Thank you for providing your Consort 2010 statement: extension to cluster randomised trials checklist. Please replace the page numbers with paragraph numbers per section (e.g. "Methods, paragraph 1"), since the page numbers of the final published paper may be different from the page numbers in the current manuscript.

SUPPLEMENT

1) eSupplement S9/eTable 6: Please remove the ‘E’ from ‘Standard Error E’.

2) eSupplement S11 Figure 1: I assume that the dots/lines for "lockdown 2021" are not meant to show values, but to represent the time period of the lockdown. I would suggest using a shaded area in the background instead of the dots/line to mark this time period so that it is not confused with the actual values displayed with the other dots and lines for 2019 and 2021.

3) In Figures eSupplement S11 Figure 1 and Figure 2, please show the axis beginning at zero. If this is not possible, please show a break in the axis.

SOCIAL MEDIA

To help us extend the reach of your research, please provide any X (formerly known as Twitter) handle(s) that would be appropriate to tag, including your own, your coauthors’, your institution, funder, or lab. Please respond to this email with any handles you wish to be included when we tweet this paper.

Comments from Reviewers:

Reviewer #2: I am in agreement that there is no clear definition of targeted screening (DOI: 10.1016/j.lanepe.2022.100353). The design of the study with a questionnaire asking about risk factors for illness and symptoms prior to testing at the very least served as a potential barrier to broader uptake of universal screening in these EDs than simply verbally offering a test to everyone during triage. In addition, the data presented by the authors suggests that the whole process led to enrichment of individuals who had mild symptoms in the screened population. How to define what those limitations are is irrelevant, as long as they are acknowledged and they are in the submitted manuscript.

The authors have addressed many issues raised for this re-submission and I have no further concerns.

Reviewer #3: Thank you to the authors for addressing my previous comments well. I have no further issues to raise.

[LINK]

---

## [Editor Report · Decision Letter 3]

2 Nov 2023

Dear Dr Leblanc, 

On behalf of my colleagues, I am pleased to inform you that we have agreed to publish your manuscript "INTENSIFIED SCREENING FOR SARS-CoV-2 IN 18 EMERGENCY DEPARTMENTS IN THE PARIS METROPOLITAN AREA, FRANCE (DEPIST-COVID): A CLUSTER-RANDOMIZED, TWO-PERIOD, CROSSOVER TRIAL" (PMEDICINE-D-23-01467R3) in PLOS Medicine.

Prior to final acceptance, please make the following changes:

In the data statement (submission form) please briefly state the reason the data are not publicly available, e.g., for protection of participant confidentiality;

At line 25 (abstract), please remove "18";

In the abstract, around line 32, please add a sentence to describe the randomization process, e.g., "At the start of the first period, 18 EDs were randomized to the intervention or control strategy by balanced block randomization with stratification, with the alternative condition being applied in the second period.";

At line 38, we suggest "... by intention-to-treat";

At line 52, and in other places in the text where similar wording occurs, we suggest adapting the text to "In this study, we found that intensified screening ...";

In the author summary, please use the active voice in one or two places, e.g., at line 69 "We identified whether an intensified ...";

At line 402 (Discussion section), we suggest adapting the text to "... who would not otherwise have been screened ...". 

PRESS

Sincerely, 

Richard Turner PhD, for Alexandra Schaefer, PhD 

Consulting editor, PLOS Medicine

plosmedicine@plos.org